

# The carbon and nitrogen budget of *Desmophyllum dianthus*—a voracious cold-water coral thriving in an acidified Patagonian fjord

Sandra R. Maier[1,2], Carin Jantzen[1], Jürgen Laudien[1], Verena Häussermann[3,4], Günter Försterra[5], Astrid Cornils[1], Jutta Niggemann[6], Thorsten Dittmar[6,7] and Claudio Richter[1,8]

[1] Department of Biosciences, Alfred Wegener Institute Helmholtz Center for Polar and Marine Research, Bremerhaven, Germany
[2] Department of Estuarine and Delta Systems, Royal Netherlands Institute for Sea Research (NIOZ-Yerseke), Yerseke, Netherlands
[3] Facultad de Economía y Negocios, Universidad San Sebastián, Puerto Montt, Chile
[4] Huinay Foundation, Puerto Montt, Chile
[5] Escuela de Ciencias del Mar, Facultad de Recursos Naturales, Pontificia Universidad Católica de Valparaíso, Valparaíso, Chile
[6] Institute for Chemistry and Biology of the Marine Environment (ICBM), University of Oldenburg, Oldenburg, Germany
[7] Helmholtz Institute for Functional Marine Biodiversity (HIFMB), University of Oldenburg, Oldenburg, Germany
[8] Department of Biology/Chemistry, University of Bremen, Bremen, Germany

Corresponding author
Sandra R. Maier,
mail.maier.sandra@gmail.com

## ABSTRACT

In the North Patagonian fjord region, the cold-water coral (CWC) *Desmophyllum dianthus* occurs in high densities, in spite of low pH and aragonite saturation. If and how these conditions affect the energy demand of the corals is so far unknown. In a laboratory experiment, we investigated the carbon and nitrogen (C, N) budget of *D. dianthus* from Comau Fjord under three feeding scenarios: (1) live fjord zooplankton (100–2,300 μm), (2) live fjord zooplankton plus krill (>7 mm), and (3) four-day food deprivation. In closed incubations, C and N budgets were derived from the difference between C and N uptake during feeding and subsequent C and N loss through respiration, ammonium excretion, release of particulate organic carbon and nitrogen (POC, PON). Additional feeding with krill significantly increased coral respiration (35%), excretion (131%), and POC release (67%) compared to feeding on zooplankton only. Nevertheless, the higher C and N losses were overcompensated by the threefold higher C and N uptake, indicating a high assimilation and growth efficiency for the krill plus zooplankton diet. In contrast, short food deprivation caused a substantial reduction in respiration (59%), excretion (54%), release of POC (73%) and PON (87%) compared to feeding on zooplankton, suggesting a high potential to acclimatize to food scarcity (*e.g.*, in winter). Notwithstanding, unfed corals 'lost' 2% of their tissue-C and 1.2% of their tissue-N per day in terms of metabolism and released particulate organic matter (likely mucus). To balance the C (N) losses, each *D. dianthus* polyp has to consume around 700 (400) zooplankters per day. The capture of a single, large krill individual, however, provides enough C and N to compensate daily C and N losses and grow tissue reserves, suggesting that krill plays an important nutritional role for the

fjord corals. Efficient krill and zooplankton capture, as well as dietary and metabolic flexibility, may enable *D. dianthus* to thrive under adverse environmental conditions in its fjord habitat; however, it is not known how combined anthropogenic warming, acidification and eutrophication jeopardize the energy balance of this important habitat-building species.

## INTRODUCTION

Corals are ecosystem engineers (*Jones, Lawton & Shachak, 1994*), forming reefs and other 'marine animal forests' (*Rossi et al., 2017*), that are amongst the most diverse ecosystems on Earth (*Jones & Endean, 1973*; *Henry & Roberts, 2016*). As calcifying organisms, scleractinian corals are vulnerable to anthropogenically-caused ocean acidification (*Cohen & Holcomb, 2009*). Increased atmospheric carbon dioxide dissolves in seawater, where it decreases the pH (*Caldeira & Wickett, 2003*) and the saturation state of aragonite, a crystal form of calcium carbonate (*Turley, Roberts & Guinotte, 2007*). Since corals form aragonite skeletons, ocean acidification affects their calcification and skeletal growth (*Cohen & Holcomb, 2009*; *Hennige et al., 2015*; *Büscher, Form & Riebesell, 2017*). To maintain calcification under these conditions, the corals may up-regulate their internal pH (*Trotter et al., 2011*; *Anagnostou et al., 2012*; *McCulloch et al., 2012*), but this is an energy-costly process requiring a corresponding energy supply (*Gattuso, Allemand & Frankignoulle, 1999*; *Cohen & Holcomb, 2009*). Accordingly, experimental studies have suggested that enhanced heterotrophic feeding could partially counteract the negative impact of ocean acidification on coral calcification (*Cohen & Holcomb, 2009*; *Georgian et al., 2016*; *Martínez-Dios et al., 2020*).

Cold-water corals (CWCs) form one of the most structurally-complex habitats of the deep sea (*Roberts, Wheeler & Freiwald, 2006*), but occur also in shallower waters of temperate fjords (*Freiwald et al., 2004*). Since aragonite saturation is lower in the deep and cold (*Chen, Feely & Gendron, 1988*; *Jiang et al., 2015*), scleractinian CWCs are considered particularly vulnerable to ocean acidification (*Turley, Roberts & Guinotte, 2007*). Most CWC ecosystems are predicted to experience aragonite undersaturation by 2100 (*Guinotte et al., 2006*; *Zheng & Cao, 2014*). Nevertheless, in the North Patagonian Comau Fjord (Los Lagos Region, Chile; Fig. 1), the scleractinian CWC species *Desmophyllum dianthus* thrives under low pH (7.4–8.4), near and below aragonite saturation ($\Omega_{aragonite}$: 0.9 to 1.6; (*Fillinger & Richter, 2013*; *Jantzen et al., 2013a*)). The low pH may relate to the high organic matter concentration from high productivity in the area (*Montero et al., 2011*) and terrestrial run-off to the fjord (*Försterra & Häussermann, 2003*; *Jantzen et al., 2013a*). The naturally low aragonite saturation state provides a rare opportunity to study the food demand of a CWC under geochemical conditions, which most CWCs will face by the end of the century (*Försterra, Häussermann & Laudien, 2016*).

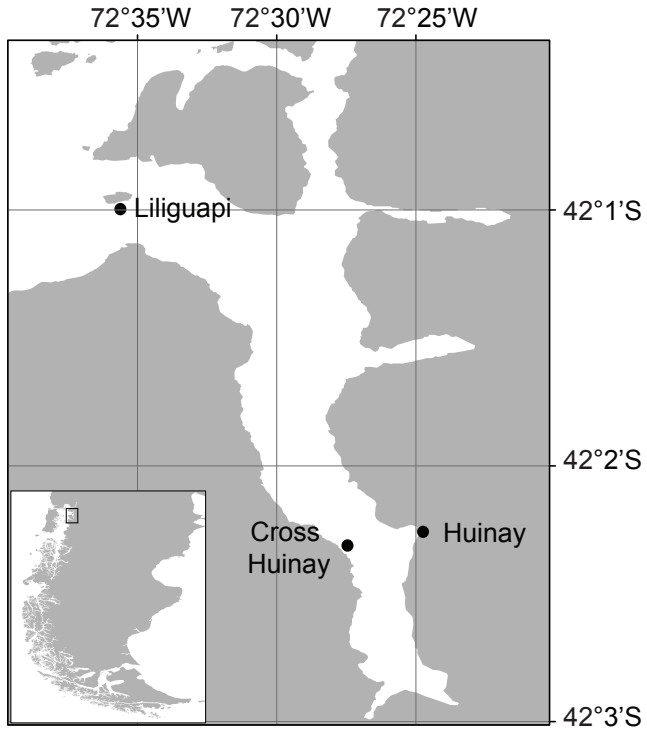

**Figure 1** **Map of Comau Fjord, located in North Patagonia, Los Lagos Region, Chile.** Shown are Huinay Scientific Field Station (Huinay) and the sampling sites Liliguapi and Cross Huinay.

*Desmophyllum dianthus* is a solitary, ahermatypic (not reef-building) CWC species with a cosmopolitan distribution (*Försterra & Häussermann, 2003*; *Cairns, Häussermann & Försterra, 2005*). In Comau Fjord, the species forms vast coral banks between 20 and 280 m water depth, particularly on steep, partly overhanging walls (*Försterra & Häussermann, 2003*; *Fillinger & Richter, 2013*; *Försterra, Häussermann & Laudien, 2016*). Their sheltered occurrence, in an upside-down position under overhangs, has been interpreted as avoidance to sedimentation (*Försterra & Häussermann, 2003*). Though ahermatypic, a diverse benthic community is (facultatively) associated with the coral banks, including sponges, bryozoans, tube-forming polychaetes, anthozoans, and bivalves (*Försterra et al., 2005*; *Försterra, Häussermann & Laudien, 2016*). Further, *D. dianthus* from the shallow parts of Comau Fjord typically has live tissue only on the apical end of the corallum (skeleton), while the basal end is bare and provides a settlement substrate (habitat) for various epibiontic and endolithic organisms, *e.g.*, foraminiferans, bio-eroding sponges, and photoautotrophic microorganisms (*Försterra et al., 2005*; *Hassenrück et al., 2013*).

The natural conditions of low pH and relatively high turbidity in the fjord region are exacerbated by climate change and intense salmon aquaculture (*Buschmann et al., 2009*; *Mayr et al., 2014*; *Iriarte, 2018*). It is therefore important to (1) know the energetic costs involved in coping with present and future fjord environments (low pH, high turbidity) and (2) estimate the resilience to disturbance affecting the energy budget, growth and

reproduction of these ecosystem engineers (*Melzner et al., 2009*; *Findlay et al., 2011*; *Vidal-Dupiol et al., 2013*). The energy budget of the fjord corals, *i.e.*, their food supply against energetic costs, is presently unknown. The carbon and nitrogen isotopic composition of *D. dianthus* from Comau Fjord indicates mostly consumption of zooplankton (*Mayr et al., 2011*). With its large polyps (up to six cm in diameter, *Försterra & Häussermann, 2003*), *D. dianthus* is able to capture not only mesozooplankton (0.2–2 mm length), such as copepods, but also larger micronekton, such as euphausiids (krill) (*Sokol, 2012*; *Höfer et al., 2018*). To fuel its respiratory carbon demand, *D. dianthus* from the Mediterranean deep sea requires the equivalent of three adult brine shrimps (*Artemia salina*) per day (*Naumann et al., 2011*). However, *A. salina* does not occur in the coral habitat and feeding on natural prey may entail a different carbon budget (*Møller & Riisgård, 2007*). In the North Patagonian fjord region, the zooplankton abundance shows a pronounced seasonality, with a maximum following the spring phytoplankton bloom and a minimum in Austral winter (*Iriarte et al., 2007*; *González et al., 2010*). In experiments, increased availability of zooplankton food enhanced the skeletal growth of the fjord corals (*Martínez-Dios et al., 2020*), but it is unknown whether food is currently limiting coral growth in the fjord region.

Here, we investigate the carbon and nitrogen (C and N) budget of the CWC *D. dianthus* under present-day low-pH conditions in Comau Fjord, Chile, *i.e.*, the difference between C and N uptake and C and N loss (Fig. 2). CWCs release the indigestible parts of their food and coral mucus (*Wild et al., 2008*) as particulate and dissolved organic matter (POM, DOM), measurable as particulate and dissolved organic carbon and nitrogen (POC, PON; DOC, DON). CWC metabolism involves oxygen consumption and carbon dioxide production through respiration, as well as ammonium production via excretion (*Khripounoff et al., 2014*; *Maier et al., 2019*). The remaining, non-released C and N is available for growth of somatic and reproductive tissue and therefore termed 'scope for growth' (*Warren & Davis, 1967*). Since C and N uptake and loss typically depend on meal size and quality (*Secor, 2008*), we determined the C and N budget (uptake versus loss) of the fjord corals under three different feeding scenarios, simulating the varying zooplankton availability in the fjord region (*Iriarte et al., 2007*; *González et al., 2010*): (1) live fjord zooplankton (100–2,300 μm), (2) live fjord zooplankton plus larger krill (>7 mm), and (3) short-term food deprivation. From these budgets, we estimated the minimum C, N and zooplankton demand of *D. dianthus* in the North Patagonian fjord region and evaluated the scope for growth of this habitat-forming CWC species.

## MATERIALS & METHODS

### Coral collection and maintenance

Thirteen similar-sized *D. dianthus* specimens (calyx height: 3.4 ± 0.4 cm, calyx length: 1.4 ± 0.2 cm, calyx width: 2 ± 0.4 cm, Fig. 3A) were collected live from 20 m water depth in Comau Fjord (Cross Huinay, Liliguapi, Fig. 1) in January 2012. The collection of *D. dianthus* for scientific purposes was approved by the Chilean Ministry of Economy, Development & Tourism, sub-secretariat of fisheries and farming (ref. 1742). The corals were chiseled off the substrate by SCUBA divers and immediately placed into water-tight,

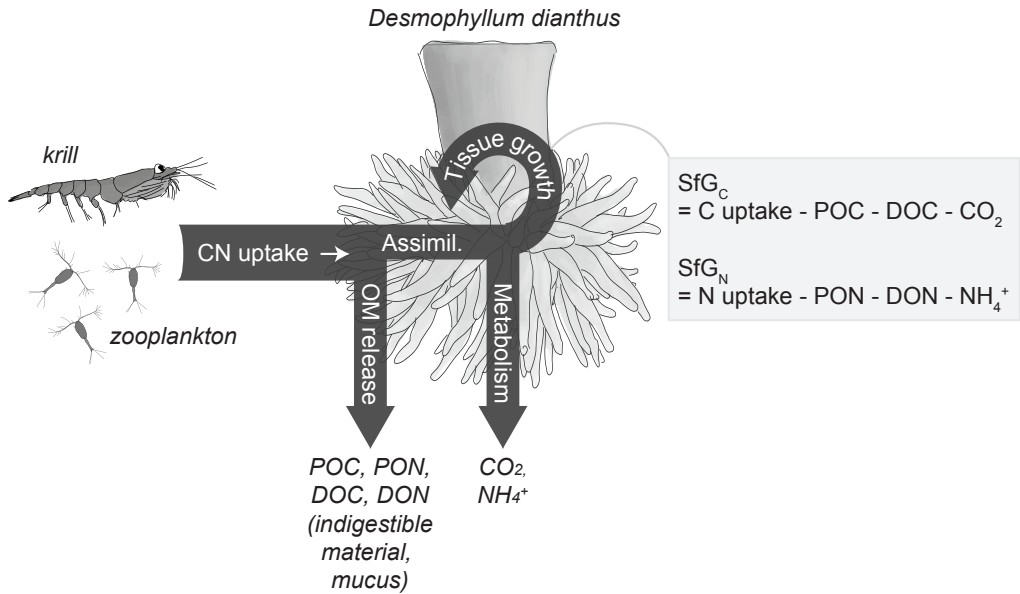

**Figure 2 Conceptual carbon (C) and nitrogen (N) budget of the cold-water coral *Desmophyllum dianthus*.** The corals release parts of the taken-up C and N as organic matter (OM), *i.e.*, particulate and dissolved organic carbon and nitrogen (POC, PON, DOC, DON), the remainder is assimilated ('Assimil.'). Parts of the assimilated C and N are lost during metabolism as carbon dioxide ($CO_2$) and ammonium ($NH_4^+$), the remainder is invested in the growth of somatic and reproductive tissue and is termed scope for growth (SfG). The figure is modified from *Soetaert & van Oevelen (2009)* and *Warren & Davis (1967)*.

sealed plastic containers, ensuring no contact with either the brackish surface water layer or air during transport to the laboratory. To remove epibiontic and endolithic organisms (*Försterra & Häussermann, 2003*), the bare corallum was carefully cut off with a submerged diamond blade (3.2 mm thick), connected to an electric grinder (*Jantzen et al., 2013b*). The fracture zone was sealed with cyano-acrylate gel (super flex glue gel) and glued to a polyethylene screw (*Jantzen et al., 2013b*), which served to fix the corals in their natural 'upside-down' growth position (Fig. 3A).

For maintenance before and during the experiment, corals were kept in three 28 L-maintenance tanks (Fig. 3A, maximum seven corals per tank) with a flow-through of 10 μm- filtered fjord water, continuously pumped from 25 m depth off Huinay (flow: 4.2 L h$^{-1}$; temperature: 11.9 °C; salinity: 31.6; particulate organic matter concentration: $12.8 \pm 4.2$ μmol POC L$^{-1}$; $1.4 \pm 1.2$ μmol PON L$^{-1}$). Before the start of the experiment, corals were fed for two hours per day with 1/4 of a haul of live, freshly-collected fjord zooplankton (see next section). This corresponds to a zooplankton concentration of $571 \pm 203$ zooplankton individuals L$^{-1}$ (mean $\pm$ standard deviation), equivalent to $85 \pm 30$ μmol C L$^{-1}$ and $16 \pm 6$ μmol N L$^{-1}$, as analyzed in four additional aliquots of 1/4 of a zooplankton haul (see section 'Sample analysis'). During feeding, water exchange was interrupted.

Before the start of the experiment, all corals were weighed in seawater of ambient temperature and salinity (density 1.026 g cm$^{-3}$) with an analytical balance equipped with

### a   Coral maintenance

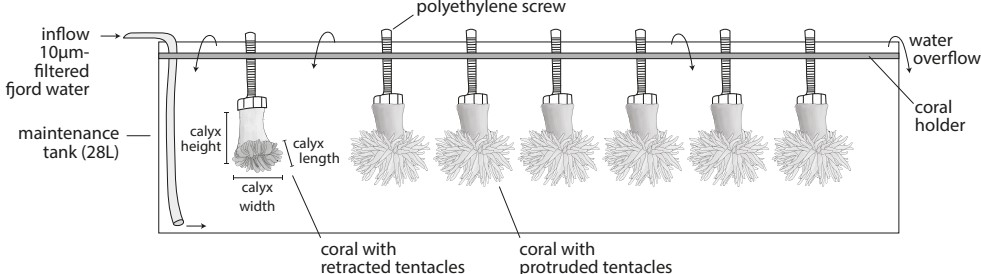

### b   Feeding treatments & measurement of CN uptake

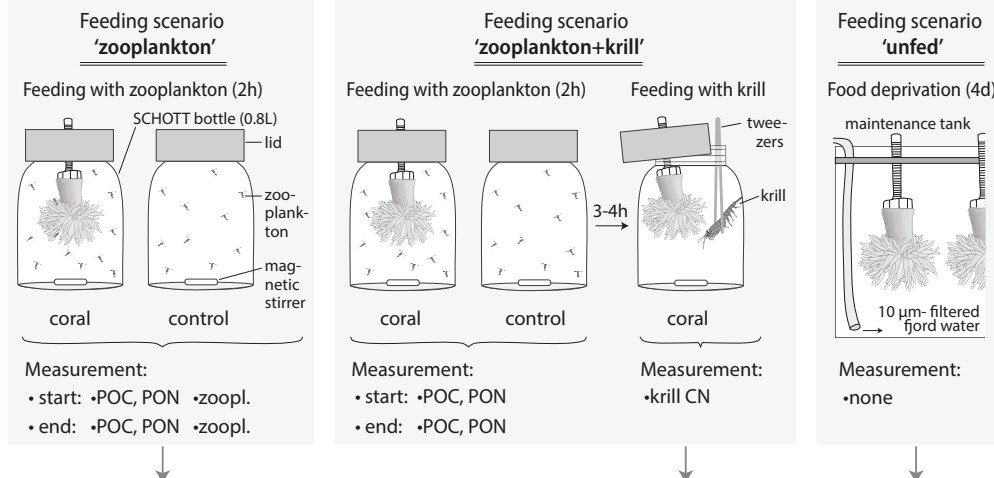

### c   Incubations to measure CN loss

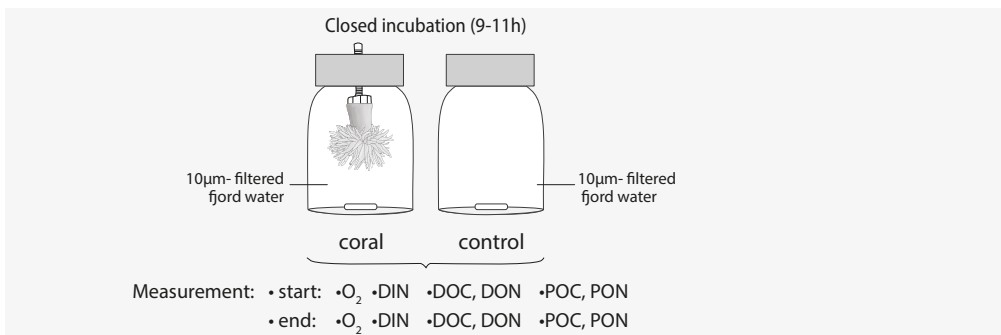

**Figure 3** **Coral maintenance, experimental set-up and measurements.** (A) Coral maintenance before and during the experiment. (B) Feeding treatments within the feeding scenarios 'zooplankton', 'zooplankton+krill', and 'unfed'. (C) Incubations following the feeding treatment in every feeding scenario. (B, C) Measurements at the start and end of the feeding/incubation are indicated. *i.e.*, 'zoopl.': number of zooplankters in SCHOTT bottle; 'POC, PON': concentration of particulate organic carbon and nitrogen; 'krill CN': carbon and nitrogen content of krill; 'O$_2$': oxygen concentration; 'DIN': concentration of dissolved inorganic nitrogen (ammonium, nitrate, nitrite); 'DOC, DON': concentration of dissolved organic carbon and nitrogen.

an underfloor weighing basket (Sartorius CPA225DOCE). Polyp dry mass was derived from buoyant weight according to *Davies (1989)*, using a species-specific aragonite density of 2.835 g cm$^{-3}$ (*Naumann et al., 2011*). Corals were given a recovery and acclimatization time of three weeks from collection and preparation to the start of the experiment.

## Zooplankton and krill collection

Live fjord zooplankton was collected ca. 1 km off the Huinay Scientific Field Station (Fig. 1) every afternoon, by a single vertical haul from 20 m water depth with a 100 µm-Nansen net (diameter 0.7 m). The size range of individual zooplankters (100–2,300 µm maximum extension) was measured under the binocular on three subsamples. To feed the corals in their maintenance tanks (previous section), the zooplankton haul was split with a Motoda plankton sample splitter into equal 1/4 portions. To feed the corals with zooplankton as part of the experimental feeding treatments (see below), the respective zooplankton haul was split into equal 1/8 portions, which were split again with measurement cylinders into ten equal portions (1/80 of the original haul).

Krill (euphausiids, *i.e.*, *Euphausia vallentini*, adult and pre-adult stages) were collected at night in Comau Fjord between 20 and 80 m water depth, in a 45-min horizontal trawl at 2 knots with a ring trawl net (0.5 m diameter, 500 µm mesh size). Krill were drained on paper tissue, measured (cephalo-thoracic length, ca. seven mm), weighed (wet mass) and stored frozen (−13 °C) until utilization in the experiment (see below). Six euphausiids were dried (40 °C, three days), weighed again (dry mass, 3.3 ± 1.5 mg) and used for C and N analysis (see below).

## Experimental design

This section focuses on the experimental design, as shown in Fig. 4, while experimental set-up, feeding and incubations are detailed in the following sections. A detailed chronology of the experiment is provided in Table S1.1.

To determine the C and N budget of *D. dianthus*, we designed a two-batch repeated-measures laboratory experiment with in total three different feeding scenarios (Fig. 4). This experimental design was chosen to simulate the temporally variable food availability in Comau Fjord (see Introduction). Both coral batches were initially offered live fjord zooplankton (Fig. 4, feeding scenario 'zooplankton'), which presumably corresponds to the natural 'baseline' food situation in the field (*Mayr et al., 2011*). Batch I ($n = 6$ corals) was subsequently offered a surplus of food, *i.e.*, zooplankton plus krill (feeding scenario 'zooplankton+krill', see below), while batch II ($n = 7$ corals) remained unfed, mimicking food (zooplankton) shortage, *e.g.*, in winter (feeding scenario 'unfed', see below). Within each batch, at least two C and N budgets were determined separately for each individual coral, by repeated measurements on the same specimens (Fig. 4). To determine the C and N budgets, we first measured the C and N uptake of each individual coral from food during the respective feeding treatments (Fig. 4). Secondly, after each feeding treatment, the corals were individually incubated without food for 9–11 h (Fig. 4), to measure their C and N loss in terms of C respiration, ammonium excretion, and release of POC, PON, DOC, DON (Fig. 2). In between the different feeding scenarios (Fig. 4), the corals were

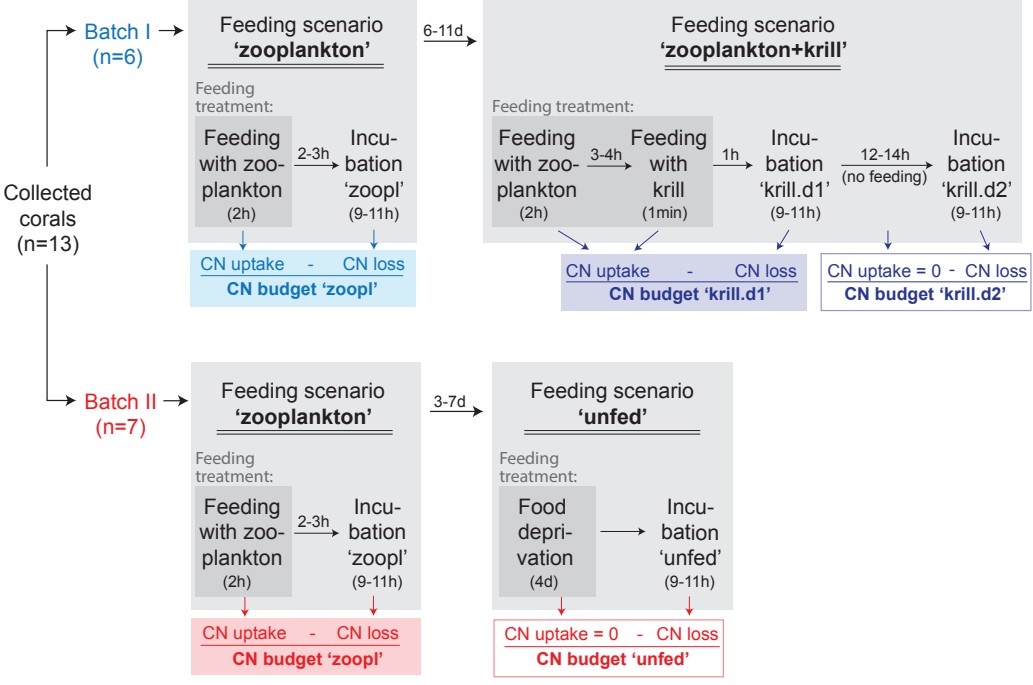

**Figure 4** **Experimental design.** The C and N budget of *Desmophyllum dianthus* was determined in a two-batch repeated-measures laboratory experiment. Each of the two coral batches was exposed to two subsequent feeding scenarios, simulating the temporally variable food availability in Comau Fjord. Each feeding scenario started with a feeding treatment, during which we determined the C and N uptake of the corals. In a subsequent incubation without food, the C and N loss of the corals was measured. In addition to the coral feeding treatments and incubations, we carried out seawater controls without corals, to account for planktonic C and N fluxes (see text). The C and N budget of the corals was calculated for each feeding scenario, as the difference between C and N uptake and C and N loss. For the feeding scenario 'zooplankton + krill', two incubations were carried out; accordingly two C and N budgets were calculated. In between the feeding scenarios, corals were fed daily with zooplankton.

kept in their maintenance tanks and fed daily with live fjord zooplankton (see section 'Coral collection and maintenance'), for re-acclimatization to natural 'baseline' feeding conditions. Ideally, experimental corals should have been kept separate throughout the entire experiment, to ensure full independence of the replicates, but this was not possible for logistic reasons. Instead, each coral batch was split into subsets of one to three corals, which were distributed over three maintenance tanks. Different coral subsets within the batches were kept in different maintenance tanks and tank position was shuffled during the experiment, to achieve a partial independence of coral replicates within the batches (Table S1.1) and to minimize the odds of a differential tank effect between the batches. Experimental feeding treatments and incubations (including measurements of C and N fluxes) were done in separate experimental bottles (see section 'Experimental set-up') to ensure independence of the results.

In the 'zooplankton' feeding scenario (Fig. 4, both batches), the corals were fed with live fjord zooplankton (1,025 ± 332 zooplankters L$^{-1}$) in separate experimental bottles for 2 h (Fig. 3B; for details, see section 'Feeding with zooplankton and krill'). Such short

pulses of high zooplankton availability may occur when a zooplankton swarm with high individual densities (*Ambler, 2002*) is advected by local currents (3–11 cm s$^{-1}$, (*Jantzen et al., 2013b*)). Zooplankton swarms are common in fjord boundaries (*Hirche, Laudien & Buchholz, 2016*). The 9–11 h-incubations in separate experimental bottles (called 'zoopl') followed 2–3 h after feeding on zooplankton (for details, see section 'Incubations'); hence, C and N losses were measured between 2–14 h after feeding, covering the time it takes tropical scleractinian corals to digest zooplankton (*Sebens et al., 1996*; digestion times of CWCs are unknown). In the 'zooplankton+krill' feeding scenario (Fig. 4), the corals were additionally fed with one individual euphausiid, 3–4 h after receiving live fjord zooplankton. Taking into account the larger food ration and the presumably prolonged digestion time, we here carried out two 9–11 h-incubations, the first starting 1 h after feeding on krill (covering the period between 1–12 h after feeding, called 'krill.d1'), the second starting 24 h after feeding on krill (covering the period between 24–35 h after feeding, called 'krill.d2'). In between the incubations 'krill.d1' and 'krill.d2', the corals remained unfed. In the 'unfed' scenario (Fig. 4), corals remained unfed in their maintenance tank (10 µm-filtered seawater; all corals in one tank) for four days. After this period, corals were incubated once for 9–11 h (incubation called 'unfed').

## Experimental set-up

Feeding with zooplankton and krill, and incubations, were carried out at 11 °C (representing *in situ* temperature) in separate experimental bottles with one coral each (Figs. 3B, 3C). As experimental bottles, we used SCHOTT-DURAN® bottles of 0.5 L specified volume, which fit a total volume of 0.8 L (including the bottleneck). The bottles were filled with 10 µm-filtered fjord water, pumped from 25 m depth off Huinay. The coral was fixed in its natural 'upside-down' position (Figs. 3B, 3C) in the custom-built bottle lid (*Jantzen et al., 2013b*). A magnetic stirring bar at the bottle floor created a circular flow of ∼1cm s$^{-1}$; to estimate this flow, we filmed the particle movement in unfiltered seawater from above, stopped the time that the circulating particles needed to cross sections of known length at different orbits of the bottle, and averaged the resulting flow velocities (distance/time). The flow was high enough to kept live zooplankton in suspension, but low enough not to bend the coral tentacles (*Sokol, 2012*).

## Feeding with zooplankton and krill

The feeding scenarios 'zooplankton' and 'zooplankton+krill' (Fig. 3B) were started by adding one aliquot of live zooplankton food (1/80 of one zooplankton haul in 50 mL filtered seawater) to each of the experimental bottles. For each zooplankton-fed coral subset (Table S1.1), five additional aliquots of live zooplankton food were prepared the same way and analyzed for the amount of POC and PON ($n = 3$) and the number of zooplankton individuals ($n = 2$) added to each experimental bottle (Fig. 3B: 'start POC, PON, zooplankton'). In the feeding scenario 'zooplankton+krill', zooplankton food aliquots were analyzed for POC and PON ($n = 3$), but not for zooplankton. Start POC and PON concentrations in the feeding scenario 'zooplankton' were 164 ± 36 µmol C L$^{-1}$ and 30 ± 7 µmol N L$^{-1}$, start POC and PON concentrations in the

feeding scenario 'zooplankton+krill' were $126 \pm 0.3$ µmol C L$^{-1}$ and $24 \pm 0.1$ µmol N L$^{-1}$. The start zooplankton concentration was $1{,}025 \pm 332$ zooplankters L$^{-1}$ in the feeding scenario 'zooplankton' (mean $\pm$ standard deviation; averaged over all replicates); the start zooplankton concentration in the feeding scenario 'zooplankton+krill' was presumably similar. Per coral subset (Table S1.1), triplicate seawater controls were prepared (Fig. 3B: 'control'); these were experimental bottles without corals, but with zooplankton food, that served to determine zooplankton loss from internal zooplankton predation and handling. During feeding, polyp activity was closely monitored, to ensure that all corals had their tentacles expanded. To end the feeding after 2 h, the corals were removed from the bottles and returned to the maintenance tanks. Control incubations (seawater only) were ended with 0.5–1 h delay (*i.e.*, $t = 2.5$–3 h), due to logistical limitations (handling time). Due to the low activity in the seawater-only controls, we expect no significant bias related to the longer duration. End water samples for the amount of POC and PON (290–370 mL) and for the number of zooplankton individuals (350–490 mL; only in feeding scenario 'zooplankton') were taken from each experimental bottle after thorough mixing (Fig. 3B, 'end POC, PON, zooplankton'). POC and PON samples were collected on pre-combusted (24 h, 500 °C), pre-weighed 0.7 µm- glass-fiber filters (GF/F) by vacuum filtration. Filters were frozen ($-13$ °C) and dried to constant mass at 40 °C. Zooplankton samples were concentrated over a 55 µm-mesh and fixed in 4%-borax-buffered formaldehyde. For feeding with krill (in the feeding scenario 'zooplankton+krill'), the corals were returned to the separate experimental bottles 3–4 h after feeding on zooplankton and received one thawed, pre-measured euphausiid via tweezers (Fig. 3B).

## Incubations

For the incubations (without food), the corals were placed into the experimental (SCHOTT) bottles, that were filled with 10 µm-filtered fjord water and closed air bubble-free with a lid (Fig. 3C). Per coral subset (Table S1.1), we additionally prepared triplicate seawater controls without corals (Fig. 3C: 'control') to determine the possible effect of nano- and picoplankton oxygen ($O_2$), C and N fluxes on our results. At the start of the incubation, the start $O_2$ concentration was measured with an optode (HQ40d; Hach, USA; resolution: 0.1 mg $O_2$ L$^{-1}$) in a separate experimental bottle $B_o$ ('start $O_2$'). Start water samples were taken by syringe from $B_o$ for analysis of dissolved inorganic nitrogen (DIN, *i.e.*, ammonium, nitrate, nitrite; 100 mL), DOC, and DON (20 mL, in triplicates). The remaining water in $B_o$ (300–600 mL) was used for POC and PON analysis. During the incubation, coral polyps remained protruded with extended tentacles. At the end of the incubation, the corals were removed and returned to their maintenance tanks. In each experimental bottle, the end $O_2$ concentration was first measured, before taking end water samples for DIN, DOC, DON, POC and PON. The end $O_2$ concentration was never below 80% of the start $O_2$ concentration, a conservative threshold to avoid effects of low-oxygen concentration on coral physiology (*Dodds et al., 2007*). DIN samples were filtered through GF/F-filters into glass vials and fixed with a concentrated mercury chloride solution (0.105 g L$^{-1}$) to prevent microbial activity. Samples for DOC and DON were filtered through GF/F-filters into glass vials (filters and vials pre-combusted, 450 °C, 12 h), acidified to pH = 2 with 32%
hydrochloric acid to avoid microbial activity and stored dark at 4 °C. Sample processing for POC and PON was described in the previous section ('Feeding').

## Coral tissue sampling

At the end of the experiment, the coral tissue was removed from the skeleton with an airbrush filled with 0.7 µm-filtered seawater and homogenized with an ultra turrax (*Jantzen et al., 2013c*). The volume of the tissue-seawater suspension was measured. Subsamples of the tissue-seawater suspension (one mL, $n = 6$ aliquots per coral) were collected on pre-combusted (500 °C, 24 h), pre-weighed GF/F-filters and dried up to constant mass at 40 °C. Coral samples were transported to Germany under CITES permit E-00427/12.

## Sample analyses

Samples of coral tissue, krill, POC and PON were weighed for dry mass and subsequently analyzed for organic carbon (OC) and organic nitrogen (ON) content on an elemental analyzer (EuroEA3000, EuroVector) with acetanilide calibration (measurement precision >99%, *i.e.*, <1% relative standard deviation for triplicate measurement of acetanilide). To measure the OC content of coral tissue (tissue-C), subsamples on GF/F filters ($n = 3$ per coral) were vapor-acidified with 12 N-hydrochloric acid prior to analysis, to remove remainders of skeletal inorganic carbon (*Hedges & Stern, 1984*). The ON content of coral tissue (tissue-N) was measured on separate subsamples ($n = 3$) without acidification. The total tissue-C and -N content of each coral was calculated by multiplying the OC and ON content measured in the subsamples (one mL) with the volume of the tissue-seawater suspension (see previous section). Samples of krill, POC and PON (from feeding and incubations) were not acidified, as previous tests revealed negligible amounts of inorganic carbon. POC concentration was determined as $POC[\frac{\mu mol\ C}{L}] = \frac{POC\ content\ GF/F\ filter}{filtered\ volume}$; PON concentration was calculated accordingly. For krill, we determined the relation between wet mass and C content (C content [µmol C euphausiid$^{-1}$] = 3.9 wet mass [mg] + 48.3; R$^2$ = 0.86) and the relation between wet mass and N content (N content [µmol N euphausiid$^{-1}$] = 0.9 · wet mass [mg] + 11.4; $R^2 = 0.87$). Zooplankton samples were counted in a Bogorov chamber under a stereo microscope. Only undamaged, non-gelatinous zooplankters were counted.

Ammonium ($NH_4^+$), nitrate and nitrite concentration was analyzed spectrophoto-metrically at the ICBM-Terramare Wilhelmshaven according to *Grasshoff, Ehrhardt & Kremling (1983)* (measurement resolution: 0.01 µmol N L$^{-1}$). DOC and total dissolved nitrogen (TDN) concentration was quantified via high-temperature catalytic oxidation (HTCO, *Sugimura & Suzuki, 1988*) on a Shimadzu TOC-VcpH analyzer, equipped with a Total Nitrogen Measuring Unit (TNM-1), with L-arginine calibration. Analytical accuracy and precision were determined by analyzing reference samples (D. Hansell, University of Miami, USA) and were higher than 95%. The datasets of DOC and TDN concentrations contained a few extreme values, possibly due to sample contamination during handling and processing. Therefore, outliers of DOC and TDN concentration were identified as values < Q1 –IQR · 1.5 and values > Q3 + IQR · 1.5, where Q1 was the first quartile of the data subset, Q3 the third quartile, and IQR the interquartile range between Q1 and Q3; data

subsets were end values of coral incubations, end values of control incubations, and start values of coral and control incubations together. Outliers were excluded and the remaining start and end values of each incubation (samples taken in triplicate) were averaged. DON concentration was obtained as DON = TDN −DIN, with DIN = ammonium + nitrate + nitrite.

## Carbon and nitrogen budget

For each feeding scenario, the C and N budgets of the corals were determined as the difference between C and N uptake, measured during the feeding treatment, and C and N loss, measured in the subsequent incubation (Figs. 2 and 4). In the feeding scenario 'zooplankton+krill', two incubations followed the feeding treatment, hence two C and N budgets were determined per coral, one covering the period 1–12 h after feeding, called 'krill.d1', the second covering the period 24–35 h after feeding, called 'krill.d2' (Fig. 4).

All C, N, and $O_2$ fluxes were calculated from the difference between the measurement at the start of the feeding or incubation and the measurement at the end of the feeding or incubation, in coral ('coral') and seawater-control ('control') trials. The (total) C uptake of the corals from zooplankton (treatments 'zooplankton', 'zooplankton+krill', in µmol C) was determined as

$$C\ uptake = \frac{[POC]_{start,coral} - [POC]_{end,coral}}{t_{coral}} - \frac{[POC]_{start,control} - [POC]_{end,control}}{t_{control}} \cdot V \cdot t_{coral},$$

where [POC] is the POC concentration, t is the feeding time and V is the water volume of the coral incubation (experimental bottle volume - coral volume). The N uptake of the corals from zooplankton was determined similarly. In the feeding treatment 'zooplankton+krill', the total C or N uptake is the sum of zooplankton-C or -N uptake and the C or N content of the krill individual provided to the respective coral; the C and N content of the individual euphausiid was estimated from its wet mass (see previous section). In the feeding treatment 'unfed', a zero C and N uptake was assumed. In all feeding treatments, the C and N uptake was treated as daily rates (*i.e.*, µmol C and N d$^{-1}$), because the corals were fed once per day only. In treatment 'zooplankton', we additionally determined the zooplankton capture of the corals, as

$$Zooplankton\ capture = \frac{zoopl_{start,coral} - zoopl_{end,coral}}{t_{coral}} - \frac{zoopl_{start,control} - zoopl_{end,control}}{t_{control}}$$
$$\cdot V \cdot t_{coral},$$

where zoopl is the number of zooplankton individuals.

Hourly respiration rates of the corals (in µmol $O_2$ h$^{-1}$) were derived as

$$Respiration(O_2) = \frac{[O_2]_{start,coral} - [O_2]_{end,coral}}{t_{coral}} - \frac{[O_2]_{start,control} - [O_2]_{end,control}}{t_{control}} \cdot V,$$

where [$O_2$] is the $O_2$ concentration. The C respiration rate was derived from the $O_2$ respiration rate, assuming a respiratory quotient $CO_2$:$O_2$ = 1; this quotient was measured for CWCs *in situ* (*Khripounoff et al., 2014*). Rates of $NH_4^+$ excretion and release of POC, PON, DOC, DON (in µmol C or N h$^{-1}$) were obtained as

$$C\ or\ N\ release = \frac{[x]_{end,coral} - [x]_{start,coral}}{t_{coral}} - \frac{[x]_{end,control} - [x]_{start,control}}{t_{control}} \cdot V,$$

where [x] is the concentration of the respective substance. It should be noted that coral C and N release may have been underestimated if coral-produced material was taken up by bacteria during the incubations, *i.e.*, by bacterioplankton passing the 10 µm-filter and/or by the coral microbiome (*Schöttner et al., 2009*). Seawater control incubations without corals cannot control for this bacterial uptake. Rates of respiration, ammonium excretion, release of POC, PON, DOC and DON were extrapolated to 24 h and standardized to coral tissue organic carbon content (tissue-C) as approximation of coral biomass. To facilitate comparability with other studies, we additionally provide C and N fluxes standardized to polyp dry mass (determined via buoyant weight) and skeletal dry mass (polyp dry mass –tissue dry mass) in Table S1.2.

The daily C and N budget of the corals (Fig. 2) was derived as

$$SfG_C = C\ uptake - C\ respiration - POC\ release,$$

and

$$SfG_N = N\ uptake - NH_4^+\ excretion - PON\ release,$$

both in µmol C (mmol tissue-C d$^{-1}$). The term 'scope for growth' (SfG, *Warren & Davis, 1967*) denotes the net C or N gains of the corals, which remain from food uptake after subtraction of all C and N losses. SfG > 0 indicates a C or N surplus, which can be invested in biomass growth (somatic and/or reproductive), while SfG < 0 indicates a C or N deficit. It should be noted that DOC and DON release were excluded from the C and N budgets, due to the high variability in DOC and DON fluxes (see below). The summed C loss, *i.e.*, C respiration + POC release, was considered as minimum C demand of the corals, the summed N loss, *i.e.*, NH$_4^+$ excretion + PON release, as their minimum N demand.

## Data analysis

Graphical and statistical analysis was performed with R (*R Core Team, 2017*). Values are given as mean ± standard deviation. Firstly, we tested whether the feeding treatments (within the different feeding scenarios, Fig. 4) had an effect on the C and N fluxes, *i.e.*, on the total C and N uptake and on rates of C respiration, NH$_4^+$ excretion, and release of POC, PON, DOC, DON. For each flux, linear mixed effect (LME) models were fitted, separately for coral batch I and II (Fig. 4), with the function lmer (R package lmerTest, *Kuznetsova, Brockhoff & Christensen, 2017*). The LME models accounted for the repeated measures on the individual corals of the two batches, using coral individuals as random effect and 'feeding treatment + incubation' (*i.e.*, 'zoopl', 'krill.d1', 'krill.d2', 'unfed') as fixed effect (*i.e.*, flux ~ treatment + (1|coral_individual)). In batch I, C and N fluxes directly (*i.e.*, within the first day) after feeding on zooplankton ('zoopl') were compared with C and N fluxes directly after feeding on zooplankton+krill ('krill.d1'); and C and N fluxes directly after feeding on zooplankton+krill ('krill.d1') were compared with C and N fluxes one day later ('krill.d2'). In batch II, C and N fluxes after feeding on zooplankton ('zoopl') were compared with C and N fluxes after four days of food deprivation ('unfed'). Detailed results of the LME models are available as Table S1.10. Secondly, C and N budgets were visualized by plotting the total C and N loss over the total C and N uptake (for all C and N

budgets combined, *i.e.*, 'zoopl', 'krill.d1', 'krill.d2', 'unfed'). The lines 'C loss = C uptake' and 'N loss = N uptake', *i.e.*, 'SfG = 0', were interpreted as the minimum C and N demand of the corals to balance their C and N loss, associated with each feeding treatment.

### Methodological limitations of this study

*Ex situ* experiments, as presented here, allow the measurement of the total C and N fluxes of a CWC, *i.e.*, its C and N budget. This measurement proves difficult *in situ*, as it typically requires a closed-off water volume. Nevertheless, *ex situ* experiments only provide specific simulations of the dynamic fjord environment (*González et al., 2010*; *Jantzen et al., 2013a*; *Iriarte, 2018*). For example, the *in situ* feeding rate of the corals is unknown. Addition of an 'artificial' experimental amount of zooplankton in the feeding treatments may have led to higher or lower than natural feeding rates. Furthermore, it cannot be excluded that *ex situ*, the corals were exposed to higher stress than *in situ*. Stress can enhance the C and N loss of CWCs, *e.g.*, through increased *ex situ* respiration (*Khripounoff et al., 2014*) and stress-induced mucus production (*Zetsche et al., 2016*). However, we minimized experimental stress, by maintaining the experimental corals in natural fjord water to assure suitable water quality, and by minimum, careful handling. Corals did not show visible signs of stress, such as visibly increased mucus release, retracted tentacles or mortality. Measured DOC and DON fluxes were highly variable (see below) and were therefore excluded from the C and N budgets, which likely caused an over- or underestimate of the coral C and N demand. As noted above, bacteria may have taken up coral-produced POC and PON (and DOC, DON) during the incubations, leading to a potential underestimate of these loss terms and the coral C and N demand.

## RESULTS

### Uptake of carbon and nitrogen

When fed with zooplankton (treatment 'zooplankton'), *D. dianthus* captured $156 \pm 74$ zooplankton individuals polyp$^{-1}$ h$^{-1}$. Given the 2 h-feeding time per day, the total zooplankton capture was $312 \pm 148$ zooplankton individuals d$^{-1}$, resulting in a C uptake of $18.8 \pm 11.5$ µmol C (mmol tissue-C)$^{-1}$ d$^{-1}$ (Fig. 5A) and a N uptake of $4 \pm 2.3$ µmol N (mmol tissue-C)$^{-1}$ d$^{-1}$ (Fig. 5B). One euphausiid in addition to zooplankton increased the C and N uptake of the corals (of batch I, treatment 'zooplankton+krill') by a factor of four ($67.9 \pm 6.7$ µmol C (mmol tissue-C)$^{-1}$ d$^{-1}$; $15.8 \pm 1.8$ µmol N (mmol tissue-C)$^{-1}$ d$^{-1}$).

### Respiration and ammonium excretion

Increasing meal size enhanced the respiration and ammonium excretion rate of *D. dianthus* (Figs. 5C, 5D), but relatively less compared to the increased C and N uptake. Within the first day after feeding on zooplankton, *D. dianthus* respired on average $27.5 \pm 5.8$ µmol O$_2$ (mmol tissue-C)$^{-1}$ d$^{-1}$ and excreted $3.2 \pm 1.3$ µmol NH$_4^+$ (mmol tissue-C)$^{-1}$ d$^{-1}$. When the corals received krill in addition to zooplankton, they showed an on average 35% higher respiration rate than when they were fed with zooplankton only (Fig. 5C: 'krill.d1' versus 'zoopl', blue colors). At the same time, their ammonium excretion rate doubled

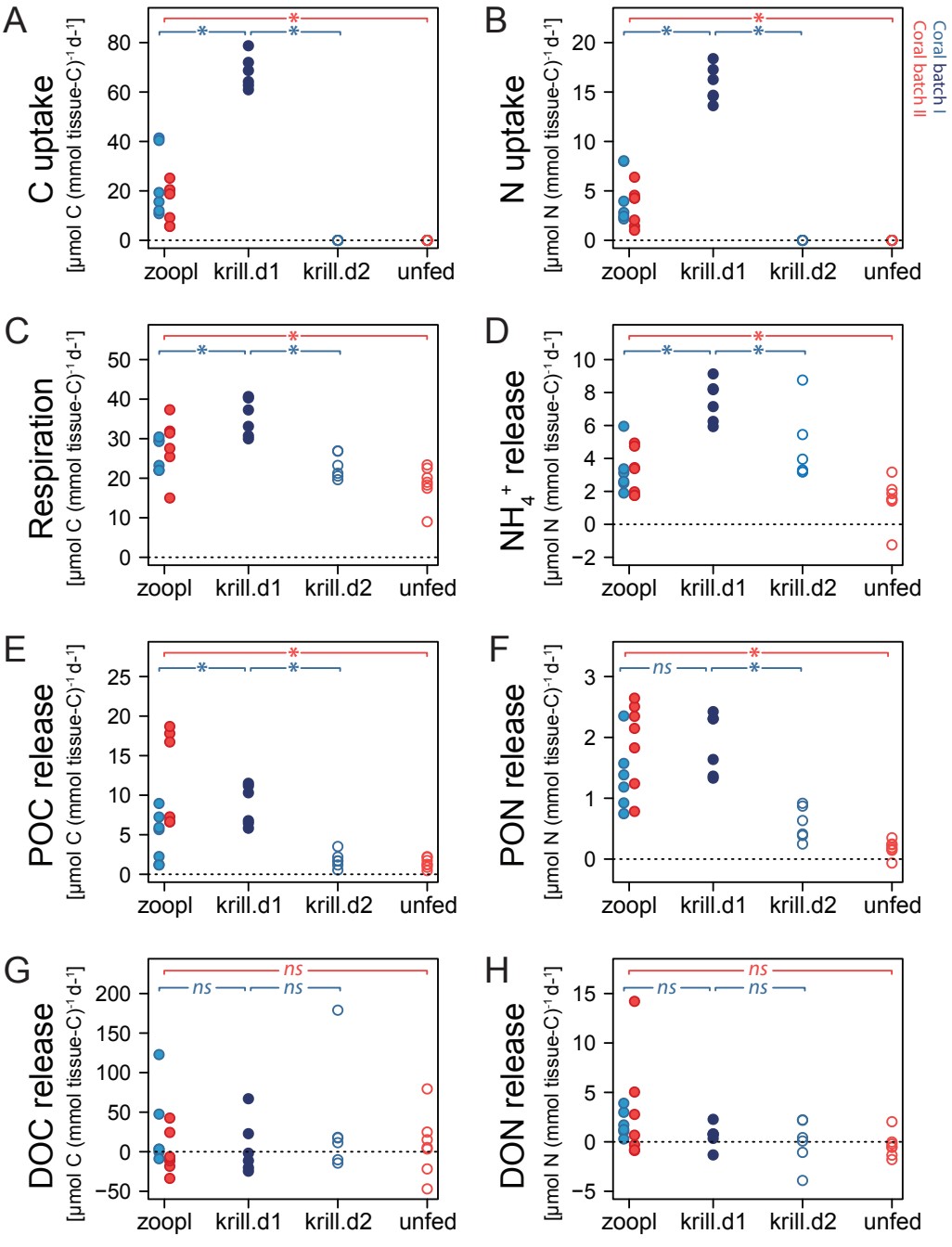

**Figure 5** **Carbon (C) and nitrogen (N) fluxes of *Desmophyllum dianthus*, exposed to different feeding treatments.** Feeding treatments were 'zoopl': fed with zooplankton; 'krill.d1': fed with zooplankton+krill, within the first day after feeding; 'krill.d2': fed with zooplankton+krill, 24 h after feeding; 'unfed': 4d-unfed. Blue colours: corals of batch I; red colours: corals of batch II. (A, B) C and N uptake during the feeding treatments; for 'unfed' and 'krill.d2', we assume no C and N uptake because corals were not fed for respectively four days and 24 h; zero values represent six (open blue circles) and seven coral replicates (open red circles). (continued on next page...)

**Figure 5 (…continued)**
(C–H) C and N loss measured in incubations after feeding: (C) respiration, (D) ammonium excretion, (E, F) release of particulate organic carbon and nitrogen (POC, PON), (G, H) release of dissolved organic carbon and nitrogen (DOC, DON). Bracket with *: linear mixed effect model found a significant difference between the indicated fluxes (see Table S1.10 for details); ns: no significant difference.

(Fig. 5D). One day after feeding on zooplankton plus krill, the corals showed a lower rate of respiration and ammonium excretion again (Figs. 5C, 5D: 'krill.d2', corals were not fed between the incubations 'krill.d1' and 'krill.d2'). Four days of food-deprivation reduced the respiration rate of the corals by 29% and their ammonium excretion rate by 54%, relative to their metabolic activity after feeding on zooplankton (Figs. 5C, 5D: 'unfed' versus 'zoopl', red colors).

### Organic matter release

*Desmophyllum dianthus* showed a clear release of POC and PON in all feeding scenarios (Figs. 5E, 5F). Directly after feeding on zooplankton, corals released $8.2 \pm 6$ µmol POC (mmol tissue-C)$^{-1}$ d$^{-1}$ and $1.7 \pm 0.7$ µmol PON (mmol tissue-C)$^{-1}$ d$^{-1}$ (Figs. 5E, 5F: 'zooplankton'). Feeding on krill in addition to zooplankton increased the POC release by 67% and the PON release by 39% compared to feeding on zooplankton only (Figs. 5E, 5F: 'krill.d1' versus 'zoopl', blue colors). One day after feeding on zooplankton plus krill ('krill.d2'), POC and PON release rates were low ($1.8 \pm 1.0$ µmol POC (mmol tissue-C)$^{-1}$ d$^{-1}$, $0.6 \pm 0.3$ µmol PON (mmol tissue-C)$^{-1}$ d$^{-1}$). Four days of food-deprivation reduced POC and PON release by 73% and 87% relative to the POC and PON release after feeding on zooplankton (Figs. 5E, 5F: 'unfed' versus 'zoopl', red colors).

DOC and DON fluxes ranged around zero with a high variability (Figs. 5G, 5H). Feeding treatments had no detectable effect. In the seawater control incubations, DOC and DON fluxes also showed a high variability with positive and negative values (Table S1.8).

### Carbon and nitrogen budget

Corals fed with zooplankton plus krill showed a positive scope for growth (SfG) for C and N within the first day after feeding ('krill.d1', Figs. 6A, 6B), meaning that their C and N uptake outweighed the combined losses via respiration, excretion, POC and PON release. Feeding on zooplankton alone did, in most cases, not provide *D. dianthus* with enough C and N for a positive SfG ('zoopl', Figs. 6A, 6B). The minimum C demand of *D. dianthus* in the experiment increased with increasing meal size (Fig. 6A, Table 1), from 20 µmol C (mmol tissue-C)$^{-1}$ d$^{-1}$ in unfed corals to 44 µmol C (mmol tissue-C)$^{-1}$ d$^{-1}$ in corals fed with zooplankton plus krill ('krill.d1'), corresponding to 52.7 to 108.4 µmol C polyp$^{-1}$ d$^{-1}$. Accordingly, the minimum N demand of *D. dianthus* (Fig. 6B, Table 1) increased from 1.7 µmol N (mmol tissue-C)$^{-1}$ d$^{-1}$ in unfed corals to 9.4 µmol N (mmol tissue-C)$^{-1}$ d$^{-1}$ in corals fed with zooplankton plus krill (krill.d1), corresponding to 3.6 to 23.8 µmol N polyp$^{-1}$ d$^{-1}$. This means that unfed corals lose 2% of their tissue-C per day to respiration and POC release and 1.2% of their tissue-N per day to ammonium excretion and PON release (Table 1). On a zooplankton diet, each polyp requires 687 zooplankters per day to balance the associated C loss (Table 1, 'zoopl'; 0.13 µmol C zooplankter$^{-1}$, Table S1.4) and

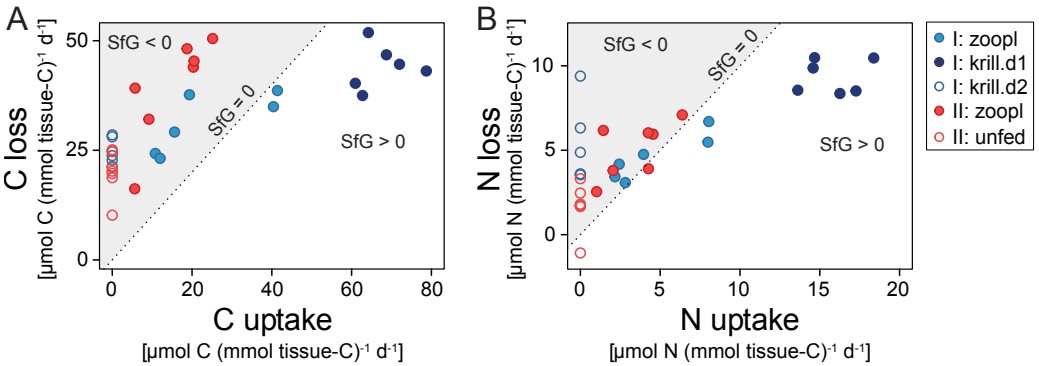

**Figure 6** **Carbon (C) and nitrogen (N) budget of *Desmophyllum dianthus* exposed to different feeding treatments, as C or N loss versus C or N uptake.** (A) C budget, (B) N budget. Dotted line: scope for growth (SfG) = 0, *i.e.*, C or N loss = C or N uptake. To the left of dotted line, marked in grey: SfG < 0, *i.e.*, C or N deficit, to the right of dotted line: SfG > 0, *i.e.*, C or N surplus. Blue colours: corals of batch I; red colours: corals of batch II. Different symbols represent the different feeding treatments, as indicated in the legend, *i.e.*, 'zoopl': fed with zooplankton, 'krill.d1': fed with zooplankton+krill, within the first day after feeding, 'krill.d2': fed with zooplankton+krill, 24 h after feeding, 'unfed': 4d-unfed. C loss includes C respiration and release of particulate organic carbon; N loss encompasses ammonium excretion and release of particulate organic nitrogen.

**Table 1** **Minimum C and N demand of *Desmophyllum dianthus*, under different feeding treatments.** Feeding treatments were 'zoopl': corals fed with zooplankton; 'krill.d1': corals fed with zooplankton+krill, within the first day after feeding; 'krill.d2': corals fed with zooplankton+krill, 24 h after feeding; 'unfed': corals 4-day unfed. The minimum C demand of the corals was derived from their C loss in terms of respiration and release of particulate organic carbon (POC); their minimum N demand was calculated from their N loss in terms of ammonium excretion and release of particulate organic nitrogen (PON). Release of dissolved organic carbon and nitrogen (DOC, DON) are excluded here. C and N demand are given in different units, as indicated.

| Feeding treatment | Min. C demand (= C loss; excluding DOC release) | | | Min. N demand (= N loss; excluding DON release) | | |
|---|---|---|---|---|---|---|
| | ($\mu$mol C polyp$^{-1}$ d$^{-1}$) | ($\mu$mol C mmol-tissue-C$^{-1}$ d$^{-1}$) | (% of tissue-C) | ($\mu$mol N polyp$^{-1}$ d$^{-1}$) | ($\mu$mol N mmol-tissue-C$^{-1}$ d$^{-1}$) | (% of tissue-N) |
| zoopl | $89.3 \pm 31.4$ | $35.6 \pm 10.3$ | $3.6 \pm 1.0$ | $11.9 \pm 3.4$ | $4.9 \pm 1.5$ | $3.3 \pm 0.9$ |
| krill.d1 | $108.4 \pm 27.2$ | $44.0 \pm 5.0$ | $4.4 \pm 0.5$ | $23.8 \pm 8.9$ | $9.4 \pm 1.0$ | $6.4 \pm 1.0$ |
| krill.d2 | $61.6 \pm 16.8$ | $24.9 \pm 2.9$ | $2.5 \pm 0.3$ | $13.5 \pm 7.5$ | $5.2 \pm 2.3$ | $3.6 \pm 1.9$ |
| unfed | $52.7 \pm 23.6$ | $20.0 \pm 4.9$ | $2.0 \pm 0.5$ | $3.6 \pm 4.1$ | $1.7 \pm 1.3$ | $1.2 \pm 0.9$ |

398 zooplankters per day to balance the associated N loss (0.03 $\mu$mol N zooplankter$^{-1}$, Table S1.4). On the other hand, less than one euphausiid per polyp per day is enough to balance the total C and N loss associated with a zooplankton plus krill diet (Table 1, 'krill.d1'; 129 $\mu$mol C euphausiid$^{-1}$, 31 $\mu$mol N euphausiid$^{-1}$, Table S1.2). Please note that DOC and DON release were not considered in the shown C and N budgets, since DOC and DON fluxes were highly variable and ranged around zero (see above). Therefore, it cannot be ruled out that the C and N demand of *D. dianthus* is higher (or lower) than indicated here.

## DISCUSSION

In this study, we report C and N budgets of the CWC *D. dianthus*, an important matrix species of CWC banks in the fjords of the Los Lagos Region, under three different experimental feeding regimes (fed with live fjord zooplankton, fed with zooplankton plus krill, four-day food-deprived). We first discuss the metabolic flexibility (metabolic rate, organic matter release) of the corals in response to varying food availability. Then, we evaluate how the corals could sustain their C and N demand under the low-pH conditions in the fjord and speculate how their C and N balance might change in the future.

### Metabolic rate

The CWC *D. dianthus* shows a high metabolic flexibility in response to varying food availability, indicated by increased respiration and ammonium excretion with increasing meal size (from four-day unfed, over zooplankton, to zooplankton plus krill, Figs. 5C, 5D). The 1.5-fold higher respiration rate within the first day after ingesting a large food ration (zooplankton plus krill, incubation 'krill.d1') as compared to one day later (incubation 'krill.d2; corals not fed between the two incubations) likely reflects the 'specific dynamic action of food' (SDA, *Rubner, 1902*). The SDA describes the increased metabolic rate of animals after feeding, owing to the energetic expenses of food capture and digestion (*McCue, 2006*; *Secor, 2008*). The SDA typically increases with meal size, but also varies with meal type (*Secor, 2008*; present study). Accordingly, krill in addition to zooplankton supplied the corals with four times more C and N compared to zooplankton only, but only increased the respiration by a factor of 1.4, probably due to higher energetic costs to capture and process many live, small zooplankters compared to one large, dead euphausiid. Energy costs to process live krill might be higher than for dead krill; however, we assume that the difference is minor, because in feeding trials with live krill, we observed that the corals immobilized their prey within a fraction of a second.

*Desmophyllum dianthus* responds to short-term food deprivation with a reduction of metabolic rates, likely to conserve energy (*Naumann et al., 2011*, present study). Corals from Comau Fjord lowered their respiration by 40% after four days of food deprivation (present study), conspecifics from the Mediterranean deep-sea by 20% after one week and by 50% after three weeks of food deprivation (*Naumann et al., 2011*). This fast, strong reduction of metabolic activity stands in contrast to the closely related CWC *Lophelia pertusa*, which reduced oxygen consumption only after several months of food deprivation (*Larsson, Lundälv & van Oevelen, 2013*; *Maier et al., 2019*). Overall, in the present study, the respiration rate of *D. dianthus* was around 2.5 times lower ($14 \pm 5$ µmol $O_2$ (g skeletal dry mass)$^{-1}$ d$^{-1}$) compared to the respiration rate of Mediterranean *D. dianthus* ($35 \pm 8$ µmol $O_2$ (g skeletal dry mass)$^{-1}$ d$^{-1}$; (*Naumann et al., 2011*)), in spite of a comparable feeding history (*Artemia salina*, krill, >24 h after feeding) and temperature (12 °C).

The metabolic flexibility of *D. dianthus* could allow colonization of different habitats, from shallow areas of fjords with a high variability of food, temperature, salinity and pH, to the environmentally more stable, deep CWC habitats (*Freiwald et al., 2004*). Global warming increases the metabolic rate of CWCs (*Dodds et al., 2007*; *Gori et al., 2016*;

*Dorey et al., 2020*), which could cause severe energetical constraints for this and other CWC species.

## Organic matter release

Cold- and warm-water corals release organic matter (POM, DOM), as feces egested from their gastrovascular cavity (*Yonge, 1930*) and as coral mucus (*Brown & Bythell, 2005*; *Naumann et al., 2011*). The high POC and PON release by *D. dianthus* within the first day after feeding (Figs. 5E, 5F) suggests that fecal material accounts for most of its POM release. The CWC species engulfs its prey whole; hence, sloppy feeding *sensu Banse (1992)*, *i.e.*, the loss of organic matter in front of the mouth, can be ruled out. Krill are assimilated at a higher efficiency compared to zooplankton (CN assimilation:CN uptake), due to the lower fecal loss in relation to the high C and N uptake (Fig. 5).

Four-day-unfed corals continue to release smaller amounts of POM, likely as mucoid material. Coral mucus consists of glycoproteins (*Bythell & Wild, 2011*) and serves as protection against sediment smothering, biofouling and as feeding aid (*Brown & Bythell, 2005*). *Wild et al. (2008)* reported that mucus of the CWC *L. pertusa* rapidly dissolved in seawater, hence, the DOC release of this CWC species was >30 times higher than its POC release. This high DOC:POC ratio was not confirmed for *D. dianthus* (present study); instead, rates of DOC release/uptake were highly variable (Fig. 5G).

Four-day unfed *D. dianthus* from Comau Fjord released on average four times more total organic carbon (7.1 $\mu$mol TOC (g skeletal dry mass)$^{-1}$ d$^{-1}$; POC plus DOC) than its Mediterranean conspecifics (1.9 $\mu$mol TOC (g skeletal dry mass)$^{-1}$ d$^{-1}$, (*Naumann et al., 2011*)). Higher mucus production in the shallow areas of the fjords may serve as protection against higher particle loads (*Larsson et al., 2013*; *Zetsche et al., 2016*). In the Comau Fjord, the chlorophyll-a concentration (up to >50 mg m$^{-3}$, *Garcia-Herrera et al., 2021*), and hence turbidity, is orders of magnitude higher than in the deep parts of the oligotrophic Mediterranean (<2 mg m$^{-3}$; *Lo Iacono et al., 2019*).

Due to relatively high mucus and fecal loss, the TOC release already contributes 30–60% to the total C loss of the fjord corals (respiration plus organic matter release). Anthropogenically increased sedimentation, *e.g.*, from the extensive salmon farming in the Chilean fjords (*Häussermann et al., 2013*; *Försterra, Häussermann & Laudien, 2016*), could further increase mucus production and related energy expenditure.

On the steep, partly overhanging fjord walls, *D. dianthus* often co-occurs with other suspension feeders, such as the bivalve *Acesta patagonica* and the sponge *Mycale thielei* (*Försterra et al., 2005*). Like *D. dianthus*, these genera produce large amounts of detrital and/or fecal material (*Maier et al., 2020b*). The organic matter 'waste' of the suspension feeding community may serve as food for detritivores living underneath the vertical or overhanging walls, ensuring a close material recycling, as suggested for *L. pertusa* reefs (*Rix et al., 2016*; *Maier et al., 2020b*).

## Carbon and nitrogen budget

The CWC *D. dianthus* in Comau Fjord is characterized by a high C and N demand. To account for their costs of respiration, ammonium excretion and POM (mucus) release,

unfed corals have to expend 2% of their tissue-C and 1.2% of their tissue-N per day. With growing meal size, feeding-related costs and losses increase the total C and N loss to 4.4% of the tissue-C and 6.4% of the tissue-N. To grow somatic and reproductive tissue, the corals need additional resources. Conspecifics from the Mediterranean show an even higher C demand, due to their higher respiration rate (*Naumann et al., 2011*). In comparison, *L. pertusa* from a Norwegian fjord has a ca. 10-times lower C demand, due to the lower respiration under the colder (8 °C), less-acidic conditions, and due to the lower POC release (*Maier et al., 2019*). Higher temperatures increase the respiration of CWCs and hence their energy demand (*Dodds et al., 2007*; *Gori et al., 2016*; *Dorey et al., 2020*). Similarly, a low pH stimulates the production of respiratory-chain enzymes in *D. dianthus*, likewise indicating an increased metabolic activity and energy demand (*Carreiro-Silva et al., 2014*). Nevertheless, the comparatively high skeletal growth rate of *D. dianthus* from Comau Fjord, in spite of (relatively) high temperatures and low pH (*Jantzen et al., 2013a*; *Jantzen et al., 2013b*), could indicate that the fjord corals are not food-limited.

To balance C (or N) losses, a medium-sized *D. dianthus* polyp in Comau Fjord has to capture almost 700 (or 400) zooplankton individuals (>100 μm) per day. As a voracious zooplankton predator, *D. dianthus* is capable to exploit high concentrations of zooplankton (*Höfer et al., 2018*), which may occur in swarms at densities of >1,000 zooplankters L$^{-1}$ (*Ambler, 2002*). Zooplankton aggregation was observed at pycnoclines (*Tiselius, Nielsen & Nielsen, 1994*), near oceanographic fronts, and in the vicinity of abrupt topography like seamounts or coral reefs (*Genin et al., 1994*). These aggregations form because zooplankton actively maintains its depth by swimming against vertical currents (*Genin et al., 2005*). A surprising finding in our experiments was that simulated zooplankton swarms of >1,000 individuals L$^{-1}$, leading to the capture of >300 zooplankters, were insufficient to balance the daily C and N losses of the fjord corals.

This suggests that krill, or other larger prey, play a crucial role for the nutrition of *D. dianthus* in Comau Fjord. The capture of one euphausiid alone boosts the C and N budget of the corals (Fig. 6), firstly due to its ca. 1,000 times higher C and N content compared to small zooplankton. Secondly, krill is processed more efficiently compared to zooplankton: On a krill plus zooplankton diet, corals showed a higher assimilation efficiency (lower feeding-related POC and PON release in relation to C and N uptake, see above) and growth efficiency (lower metabolic costs, *i.e.*, respiration, ammonium excretion in relation to C and N uptake, see above). We observed dense krill swarms directly next to the corals during dives by remotely-operated vehicle between 160 and 200 m depth (Fig. 7) and during SCUBA dives at 20 m depth. Likewise, the recurrence of blue whales in Comau Fjord (*Försterra & Häussermann, 2012*) indicates krill aggregations, which are known to attract the large mammals in the region (*Buchan & Quiñones, 2016*). In feeding experiments, *D. dianthus* captured live krill at a similar rate (18% h$^{-1}$) as small zooplankton (*Höfer et al., 2018*); *in situ*, krill capture remains to be quantified.

A maximized energy intake is crucial considering the pronounced seasonality in North Patagonia (*Pickard, 1971*). The C and N budget of *D. dianthus* was assessed in austral summer, when high abundances of zooplankton and krill follow the spring phytoplankton bloom (*Iriarte et al., 2007*; *González et al., 2010*) and create feast conditions. The CWCs

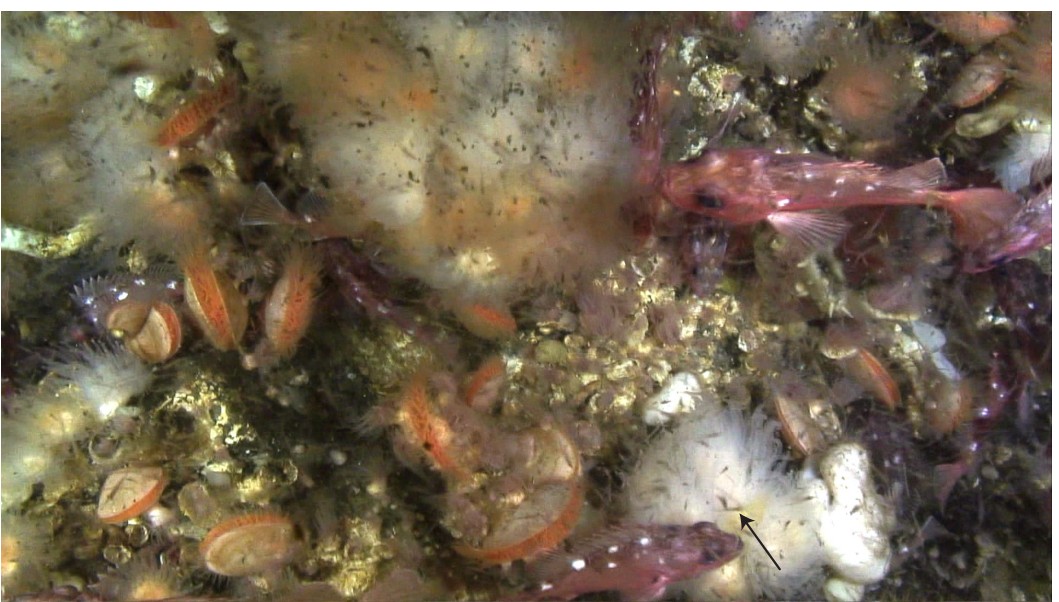

**Figure 7** **Dense swarms of krill and chaetognaths in Comau Fjord, directly above *Desmophyllum dianthus*.** Photo recorded by remotely-operated vehicle (C Richter) at 160 m depth. The arrow indicates one krill individual.

might invest the excess resources in tissue reserves such as lipids (*Maier et al., 2019*) to overcome the less-productive winter (*Iriarte et al., 2007*; *González et al., 2010*). In early spring (September), however, Patagonian *D. dianthus* also starts to produce gametes, which is an energy-costly process (*Feehan, Waller & Häussermann, 2019*). Reduced skeletal growth in summer could therefore indicate an energetic trade-off between investment in reproductive tissue and growth (*Hassenrück et al., 2013*), as suggested for *L. pertusa* from a Norwegian fjord (*Maier et al., 2020a*). In winter, when zooplankton and krill abundance is reduced, the corals may benefit from their metabolic flexibility. The fast downregulation of metabolic rate and POM release constrains C and N losses. Further, after prolonged (3-week) zooplankton exclusion, *D. dianthus* was observed to take up alternative resources, such as DOM and/or non-zooplankton POM (*Naumann et al., 2011*). A potential diet shift to more degraded material in winter was also described for the CWC *L. pertusa* (*Maier et al., 2020a*). Metabolic and dietary flexibility enable the fjord corals to survive without particulate food for several months, as we recently observed.

By the end of the century, most CWC ecosystems are predicted to face ocean acidification with anthropogenically lowered pH levels comparable to those occurring naturally in Comau Fjord at present (*Guinotte et al., 2006*; *Jantzen et al., 2013a*; *Jantzen et al., 2013b*; *Jantzen et al., 2013c*; *Fillinger & Richter, 2013*). Our study indicates that a high energy supply is crucial for *D. dianthus* to grow in its low-pH fjord habitat. Similarly, CWCs showed a higher (growth) resilience to experimentally acidified conditions when their energy supply was high (*Georgian et al., 2016*; *Martínez-Dios et al., 2020*). On top of food quantity (*e.g.*, the amount of C and N), a high food quality increases assimilation and growth

efficiency of CWCs (present study) and hence their reserves to actively counteract adverse conditions (*Carreiro-Silva et al., 2014*). Nevertheless, there are currently no indications that zooplankton and krill supply to CWC may increase to offset a changing C and N budget in the future. In contrast, on a global scale, climate change is decreasing the productivity at the ocean surface, the efficiency of the biological pump and accordingly the food supply to deeper, benthic ecosystems (*Bopp et al., 2001*; *Bopp et al., 2005*; *Gregg et al., 2003*). A decreased energy supply, in combination with an increased energy demand, could have a large, negative impact on the fitness of the local population of *D. dianthus* we studied in particular, and global CWCs in general.

## CONCLUSIONS

The CWC *D. dianthus* from the fjords of the Los Lagos Region in North Patagonia requires a substantial supply of pelagic food to balance its daily C and N loss through respiration, ammonium excretion and POM release. Experimental feeding on zooplankton alone was not enough to balance their C and N loss, despite the high zooplankton food concentration. For a balanced C (or N) budget, the solitary coral needs to capture a minimum of 700 (or 400) zooplankton individuals per polyp and day, or one larger prey item such as one euphausiid. Under experimental food deprivation, the corals swiftly reduced all C and N loss terms, likely to conserve energy. We argue that the exploitation of zooplankton swarms and/or the consumption of krill, combined with a high metabolic flexibility, are important in sustaining the energetic requirements of *D. dianthus* in Comau Fjord under naturally low pH. The bulk of the population, however, thrives in deeper waters under even lower pH, near or below aragonite saturation (*Fillinger & Richter, 2013*; *Jantzen et al., 2013a*), which may further increase their energy demand (*Gattuso, Allemand & Frankignoulle, 1999*; *Cohen & Holcomb, 2009*); this may be evaluated in the future by carrying out similar measurements as presented here *in situ*, along the pH depth gradient. Climate change, ocean acidification and the intense salmon aquaculture in the Patagonian fjord region likely impact the energetic balance of *D. dianthus*. The species appears particularly sensitive to a combination of stressors, such as high temperatures and acidification (*Gori et al., 2016*) or hypoxia and elevated levels of sulfide/methane (*Försterra et al., 2014*). A disrupted energy balance may have severe consequences for the growth, reproduction and hence the distribution of this and other habitat-forming cold-water coral species.

## ACKNOWLEDGEMENTS

We thank the team of the Fundación Huinay, especially Reinhard Fitzek, Dan Genter, Soledad Gonzáles, Fernando Hernández, Mauri Melipillán, and Emma Plotnek and for logistic support, and Lisa Reichel and Jens Müller for Scientific Diving. Gerd Liebezeit from the ICBM-Terramare Wilhelmshaven provided nutrient analysis. Matthias Friebe kindly assisted in DOC analysis. This is publication no. 172 of Huinay Scientific Field Station.

### Funding
The study was funded by the German Federal Ministry for Research and Education through the project "ERRINA" to Jürgen Laudien and Claudio Richter ("B project CHL08/001"), by AWI ("PACES T1WP6") to Sandra R Maier, Carin Jantzen, Jürgen Laudien and Claudio Richter; "Changing Earth –Sustaining our Future, Subtopics 6.1 and 4.2" to Astrid Cornils, Jürgen Laudien and Claudio Richter), by the National Fund for Scientific and Technological Development (FONDECYT projects 1150843 and 1201717) to Günter Försterra and Verena Häussermann, and by a Brede-Foundation travel grant to Sandra R Maier. The funders had no role in study design, data collection and analysis, decision to publish, or preparation of the manuscript.

### Grant Disclosures
The following grant information was disclosed by the authors:
German Federal Ministry for Research and Education: IB project CHL08/001.
AWI: PACES T1WP6.
Changing Earth –Sustaining our Future, Subtopics 6.1 and 4.2.
National Fund for Scientific and Technological Development: 1150843, 1201717.

### Competing Interests
The authors declare there are no competing interests.

### Author Contributions
- Sandra R. Maier conceived and designed the experiments, performed the experiments, analyzed the data, prepared figures and/or tables, authored or reviewed drafts of the paper, and approved the final draft.
- Carin Jantzen and Claudio Richter conceived and designed the experiments, performed the experiments, analyzed the data, authored or reviewed drafts of the paper, and approved the final draft.
- Jürgen Laudien conceived and designed the experiments, performed the experiments, authored or reviewed drafts of the paper, and approved the final draft.
- Verena Häussermann, Günter Försterra, Jutta Niggemann and Thorsten Dittmar performed the experiments, authored or reviewed drafts of the paper, and approved the final draft.
- Astrid Cornils conceived and designed the experiments, authored or reviewed drafts of the paper, and approved the final draft.

### Field Study Permissions
The following information was supplied relating to field study approvals (*i.e.*, approving body and any reference numbers):

The collection of *Desmophyllum dianthus* for scientific purposes at the indicated sampling sites was approved by the sub-secretariat of fisheries and farming within the Chilean Ministry of Economy, Development & Tourism (ref. 1742).

The coral samples were transported to Germany under CITES permit E-00427/12.
## Data Availability

All raw data and detailed statistical results are available at https://doi.org/10.5281/zenodo.5116453 and as Table S1.

## Supplemental Information

Supplemental information for this article can be found online at http://dx.doi.org/10.7717/peerj.12609#supplemental-information.

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
