# Peer review of "The carbon and nitrogen budget of Desmophyllum dianthus—a voracious cold-water coral thriving in an acidified Patagonian fjord"

_PeerJ, doi:10.7717/peerj.12609_

## Round 0.1 · original submission · Minor Revisions

This is a very well written manuscript with some interesting findings. I agree with the reviewers that minor revisions are necessary, but overall it is a worthy submission.

I would like to see the manuscript framed in a more general way. The introduction starts with CWC and quickly reduces scope to the Patagonian fjords in the second sentence. It presents important information that is relevant to ALL corals, not just CWC. I think that the broad readership of PeerJ would be attracted to a paper that is a bit more general and widely applicable.

In the entire manuscript, the presentation of the direct comparisons of respiration, excretion, and POC release are a bit misleading. If you feed them more, they will excrete more. Your finding is much more interesting than this. When they are fed krill, they increase the proportion of the C and N in their diet that they can assimilate. You should also be explicit under what conditions they are consuming 4% of their tissue C and N - when fed zooplankton only or when unfed?

I think that a clear definition of "zooplankton" would be helpful. when I first read this, I thought "Aren't krill considered zooplankton?" There is a clear size difference, and this should be easy to define. Also, please note that the word "krill" is plural.

I found the methods a bit confusing. I would rather that the feeding treatments were explained upfront, before you present the C and N losses of the corals. The figure is very helpful, but this information should be presented in the order that it is obtained in the experiment.

I hate doing this, but please consider if the results from Georgian et al. 2016 in Marine Ecology are relevant to this study. Yes, it is from my lab, but no, I am not just suggesting it so I can be cited one more time.

Finally, in the discussion, you should frame your results in the context of global climate change. Will increased feeding compensate for temperature and OA stress? Does it have to be different food, like krill, in addition to more food? Does that have implications for feeding behavior in shallow-water corals? If presented in this way, I believe that this manuscript will have a much higher impact.

Congratulations on a very nice study. I look forward to your revisions and to the publication on this manuscript in PeerJ.

·

Basic reporting

- This article uses clear and unambiguous, professional English throughout.
- There is sufficient field background/context provided in literature references.
- This article has a professional article structure, figures, tables. Raw data are shared.
- It is self-contained with relevant results to hypotheses

Experimental design

This article meets the Peer J criterias for publication regarding experimental design, see all comments in Section 4.

Validity of the findings

This article meets the Peer J criterias for publication regarding the validity of the findings, see all comments in Section 4.

Additional comments

In this manuscript, the authors investigate the carbon and nitrogen budget of a solitary cold water coral species living in environments that are usually undersaturated with respect to aragonite. These questions are of general relevance to the marine scientific community, in particular regarding the ability of these organisms to maintain themselves in future acidifying waters. This paper represents an impressive body of work and adds meaningful and needed data to the current literature regarding an azooxanthellate corals C and N metabolism.

- Introduction: The problematic is clearly presented and the introduction contains the background information necessary to understand the rationale for this study. I would however like to see explicit pH and omega values encountered in the fjords these corals live in.

- Material and methods: The methods are a bit difficult to read and follow, however the figures help greatly at understanding the design. I have added a few comments directly in the text regarding that section for things I did not feel were very clear. I think the authors should proofread this part as if they were an external reader that doesn't know this experiment to check that this part is easily understood.

- I did not understand in what the two coral batches were different? Unfortunately, I am not sure I understood if the corals were re-used for different treatments, and if that was the case, how much time there was between these feeding/incubation treatments.

- I think the material and methods would be much easier to read as a whole if there was a starting paragraph on the overall design of the experiment. Something that would go something like:
"To understand our problematic (...) we designed an experiment in two (?) parts. Firstly, we performed a 1-3h feeding experiment where we used a subset of the corals we collected in Huinay fjord and fed them with either live zooplankton collected in the area or live zooplantkon combined with thawed krill (while another subset of corals were incubated without food for comparison). By measuring POC and PON at the start and end of the feeding, we calculated the particulate C and N fluxes. In a second part, we performed 9-11h incubations without food, to calculate respiration and particulate and dissovled C and N fluxes.... " An overview first would really be great before the detailed information. I for example had problems following after how much time the incubations experiments took place after feeding?

- One figure that ties the three Figures 3 together instead would be great. (for example adding arrows from Collection in Figure 2 to Figure 3A annotated with "1 week", and then -- 12-24h --> Figure 3B). This could also use different gray shadowing for different coral batches if necessary...

- It would be good that the explanatory figures are used in the text as many times as necessary to allow the reader to follow the design. For example, Line179 mentions Fig. 3a but I guess this is Figure 3B we should look at? Similar with Line 204.
- In Figure 3B, I would recommend a change from t0 and t1 to h0 and h+2 instead so that it is cleared what the (same with Figure C with h0 and h+10). It also would lift the doubt that t0 of the feeding experiment is the same as t0 of the incubation experiment.

- Line 299-300: Is this true that a good approximation of the O2:C respiration coefficient is =1? Isn't there some metabolic DIC that should be retained for calcification? Or some created by carbonate dissolution? Reading the article cited, it seems to me that the authors should have chosen the night rate of 1.5 instead? (nights are likely more representative of what happens in a coral without photosynthetic symbionths? But not exactly of course since night and day omega conditions aren't similar...). I understand it must be difficult to get a proper respiratory coefficient, however, I think the authors should add a mention that 1) the citation refers to a tropical colonial/symbiothic (?) corals and 2) there is a deviation margin around the 1 (more or less 1) depending on night/day in these and 3) there is a margin of error in these calculations of the C respiration rate, and thus the SfGc is also subject to a margin of error (maybe quantify how much the SfGc would change if they took an alternative plausible scenario of the respiratory coefficient O2:C, maybe it wouldn't change so much if C respiration does not represent a big part of the C uptake?).

- The statistic methods used seem appropriate regarding the design of this experiment.

- Results are well written and the figures presented are of high quality. I have left some small comments directly in the pdf file.

- While Figure 4 was easy to read and follow, the figure 5 was slightly confusing, as it shifts colors from the previous figure (but uses the same data). It would be best if the main blue used for Batch I was the same as the one used for Figure 4 to improve consistency.

- Discussion is well written and informative. I have very little comments on that part, they can be found in the pdf file.

·

Basic reporting

The manuscript “The carbon and nitrogen budget of Desmophyllum dianthus – a voracious cold-water coral thriving in an acidified Patagonian fjord” by Sandra Maier, Carin Jantzen, Jürgen Laudien, Vreni Häussermann, Günter Försterra, Astrid Cornils, Jutta Niggemann, Thorsten Dittmar, and Claudio Richter is an excellent, interesting, well performed and well written study determining the C and N budgets for the CWC Desmophyllum dianthus.

I have just three major comments on the experimental design, and a very few minor additional comments, that I hope can help authors to further improve their already very good manuscript.

Experimental design

(1) The study suffers for a lack of proper replication since only two batches were established, one for fed and one for unfed corals. This means that corals inside the same batch were really not as independent as compared to those in the other batch. However, this is a limitation often occurring in experiments, especially when specific conditions need to be maintained (such as for CWC). I do not think this was really affecting the results obtained (it is my personal opinion), but I think authors should at least briefly discuss this point.

(2) It is not fully clear to me if enough time was given to corals to adjust their metabolism to the two different feeding treatments before to measure their C and N budgets. May this explain the reduced respiration in the krill-fed corals? May authors comments on this?

(3) Do authors think bacteria could have played a role in the respiration of particulate and dissolved matter, so reducing the measured outputs? May authors comments on this?

Validity of the findings

no comment

Additional comments

Line 109 I suggest removing “important foundation” from this sentence. The solitary nature of Desmophyllum dianthus allow to create a dense and complex benthic community, however with limited 3D extension (as it can be seen in figure 6). Moreover, is there any evidence that the presence of Desmophyllum dianthus is strictly required for the other species in the benthic community to grow (i.e., is a foundation species)?

Line 137 A density of 2.78 is reported in Movilla et al. 2015. What is the criterium followed to choose one instead of the other?

Lines 1676-168 Was the volume 0.5 or 0.8 L? It is unclear to me.

Line 171-172 Please, specify how was the 1 cm/s flow measured.

Line 225 Should not be 0.45 filters to be used for DIC and DIN?

Line 240 I suggest changing “enumerated” to “counted”.

---

## Round 0.2 · accepted · Accept

Thank you for your response to the reviews and my comments. I believe that the manuscript is significantly improved and I am happy to accept it for publication. Congratulations.